# Acylation of agricultural protein biomass yields biodegradable superabsorbent plastics

Antonio J. Capezza [1,2✉], Faraz Muneer [2], Thomas Prade [3], William R. Newson [2], Oisik Das[4], Malin Lundman[5], Richard T. Olsson[1], Mikael S. Hedenqvist[1] & Eva Johansson [2✉]

Superabsorbent polymers (SAP) are a central component of hygiene and medical products requiring high liquid swelling, but these SAP are commonly derived from petroleum resources. Here, we show that sustainable and biodegradable SAP can be produced by acylation of the agricultural potato protein side-stream (PPC) with a non-toxic dianhydride (EDTAD). Treatment of the PPC yields a material with a water swelling capacity of ca. 2400%, which is ten times greater than the untreated PPC. Acylation was also performed on waste potato fruit juice (PFJ), i.e. before the industrial treatment to precipitate the PPC. The use of PFJ for the acylation implies a saving of 320 000 tons as $CO_2$ in greenhouse gas emissions per year by avoiding the industrial drying of the PFJ to obtain the PPC. The acylated PPC shows biodegradation and resistance to mould growth. The possibilities to produce a biodegradable SAP from the PPC allows for future fabrication of environment-friendly and disposable daily-care products, e.g. diapers and sanitary pads.

[1] Fibre and Polymer Technology Department, KTH Royal Institute of Technology, Stockholm, Sweden. [2] Plant Breeding Department, SLU Alnarp, Lomma, Sweden. [3] Biosystems and Technology Department, SLU Alnarp, Lomma, Sweden. [4] Department of Civil, Environmental and Natural Resources Engineering, Structural and Fire Engineering Division, Luleå University of Technology, Luleå, Sweden. [5] Essity Hygiene and Health AB, Gothenburg, Sweden. ✉email: ajcv@kth.se; eva.johansson@slu.se

Utilisation of readily available renewable feedstocks for material production is a more sustainable and desirable route to compete with petroleum-based materials[1–12]. Among the renewable feedstocks, proteins obtained as side-streams from industrial processes are acknowledged as a promising source to replace fossil-based materials, which are contributing to negative environmental impact[1,5–8,13–17]. Today, one of the largest underutilised agro-industrial protein side-stream is the potato protein concentrate (PPC), obtained after starch extraction[8,15,16,18]. The PPC is produced through acidification and harsh thermal processing of the downstream liquid (potato fruit juice, PFJ) from starch extraction, leading to intensive coagulation and recovery of the protein[19–21]. Global cultivation resulted in the production of more than 360 million tons of potato in 2018[22], and ~3.7 million tons of potato starch were extracted, resulting in a production of ca. 200 000 tons of PPC from ca. 10 million cubic metres of PFJ[23,24]. An environmentally attractive and useful approach that utilises renewable biomass would be to employ PPC in the fabrication of functional alternatives to petroleum-based plastics, contributing to industrial circularity and sustainability. Recent advances in the use of PPC have focused on its foaming and emulsifying properties[25–27], fabrication of bio-based plastics (using glycerol as plasticiser)[4,8,28], and synthesis of protein nanofibrils[29].

Synthetic superabsorbent polymers (SAPs) are petroleum-based polymeric networks capable of absorbing more than $10–1000 \, g \, g^{-1}$ (water), which represent the core technology of many daily-care products of single-use[30–32]. The global daily-care business market size was 130 billion USD in 2019, and the convenience of the products has led to growing popularity and an exponential increase in the marketplace[33]. To reduce the severe environmental impact of the daily-care industry, opportunities for producing natural and biodegradable SAPs based on cellulose extracted from the forest industry and/or polysaccharides grafted with acrylic acid oligomers have been investigated[34–40]. Proteins obtained as an industrial side-stream are considered advantageous raw materials for producing bio-based and biodegradable SAPs. Different protein industrial streams, e.g., soy, wheat, and potato protein, have been used in the development of bio SAPs, showing highly competitive functional properties compared to synthetic and other bio-based SAPs[18,41–46].

To mimic the chemical structure of synthetic SAPs, the aforementioned proteins are typically functionalised by acylation using a dianhydride[32,47,48]. Previous work has shown that PPC can be acylated using different reagents, reaching liquid superabsorbent properties[18]. The harsh conditions used in the potato starch industry to coagulate the protein to produce the PPC result in extensive cross-linking and damage to the protein structure. The protein coagulation process even results in the co-extraction of toxic glycoalkaloids in levels above the threshold for human consumption[18,20,26]. However, this cross-linking of PPC is beneficial for fabricating liquid absorbent products without additional chemical cross-linking agents, while PPC does not compete with food resources.

In this article, the chemical functionalisation of PPC (acylation) without using toxic functionalisation or cross-linking reagents to produce completely biodegradable superabsorbents is reported. To evaluate the possibility of implementing the acylation process before the harsh industrial coagulation of the potato protein, functionalisation was also assessed on the PFJ (containing the soluble potato protein). Acylation of the PPC with ethylenediaminetetraacetic dianhydride (EDTAD) resulted in water, saline, and blood swelling capacities that are of interest for hygiene applications and the highest absorbent capacity reported for potato protein. Acylation of the PFJ resulted in water absorption capacity that was ca. 850% higher than that of the unmodified protein and the reference PPC. The increased swelling obtained in the tested liquids, the fast equilibrium absorption, and the demonstrated biodegradability of the PPC absorbent materials makes them interesting for the future development of bio-based SAP from industrial side-streams. The additional possibilities of combining the chemical functionalisation upstream, i.e., before the industrial coagulation process of the potato protein (estimated to reduce the greenhouse emissions by 320 000 tons of $CO_2$/year), demonstrate a possible path for future work, utilising circular bio-economy principles for the development of bio-SAP products with low environmental impact.

## Results and discussion

**Liquid swelling capacity and retention.** Water swelling capacity was increased by functionalisation of PPC samples through acylation with EDTAD compared to the as-received PPC (Fig. 1a, route RA). The PPC/25ED had the highest water swelling capacity, with ca. 15 and $25 \, g \, g^{-1}$ at 1800 s (30 min) and 24 h, respectively. The PPC/5ED and PPC/10ED resembled PPC/Ref, and displayed an increase in swelling capacity by ca. 4 times compared to the PPC. The increase from 10 to 20 wt% of EDTAD (PPC/10ED to PPC/20ED), resulted in an increase in the swelling capacity from 8 to $11 \, g \, g^{-1}$ at 2400 s (Fig. 1b). The swelling capacity increase of 8 and 2 times was observed for the PPC/25ED and PPC/Ref compared to the PPC, respectively. Results also showed that 25 wt% EDTAD addition resulted in the highest swelling for PPC (Fig. 1b). The increase in the swelling capacity with the increment of the EDTAD content agrees with previous work suggesting that the incremental addition of EDTAD increases the number of lysine-residues that have reacted during the acylation process[32]. This increase in the acylated lysine-residues results in more carboxylic acid groups formed on the material, which increase the electrostatic repulsion promoting higher liquid swelling[1,46]. An additional PPC acylation using a 1:1 EDTAD:PPC ratio showed a similar degree of swelling as obtained for PPC/25ED. The result suggests that the available reactive lysine groups have been consumed, resulting in no swelling capacity effects by increasing the concentration of EDTAD beyond 25 wt%. A comparison of the highly swollen and stable PPC hydrogel obtained from the swelling of PPC/25ED and untreated swollen PPC is shown in Fig. 1c. The free swelling capacity (FSC) of PPC/25ED (30 min) in saline, Saline A, and defibrinated sheep blood resulted in ca. 9, 6, and $4 \, g \, g^{-1}$ absorption, respectively, which are within the range of previously reported protein-based superabsorbent materials, and the highest reported for PPC-based absorbents[1,18,41,42,49]. The PPC/25ED showed a rapid water uptake; ca. $12 \, g \, g^{-1}$ (1200% weight increase) after only 1 min swelling, which is illustrated in Supplementary Movie 1.

The PPC/25ED showed centrifuge retention capacity (CRC) values (after 30 min saline swelling and centrifugation at 250 RCF for 3 min) of 2.8 and $3 \, g \, g^{-1}$ in Saline A and defibrinated blood, respectively. The CRC results indicated that the PPC/25ED could retain ca. 47 and 75% of the Saline A and defibrinated blood, respectively, within the protein network (related to the FSC values). An identically performed CRC evaluation on a commercial synthetic SAP showed that the highly charged polyacrylic acid network has the ability to retain ca. 74% of Saline A and 66% of defibrinated blood (related to the FSC result). Thus, FSC in Saline A and defibrinated sheep blood of the PPC/25ED is only 28 and 20% of what was obtained for the commercial SAP. However, the rapid FSC and liquid retention ratio of the PPC/25ED were of similar magnitudes as for the commercial SAPs. Hence, the promising FSC and CRC results pave the way for the utilisation of industrial PPC side-stream to

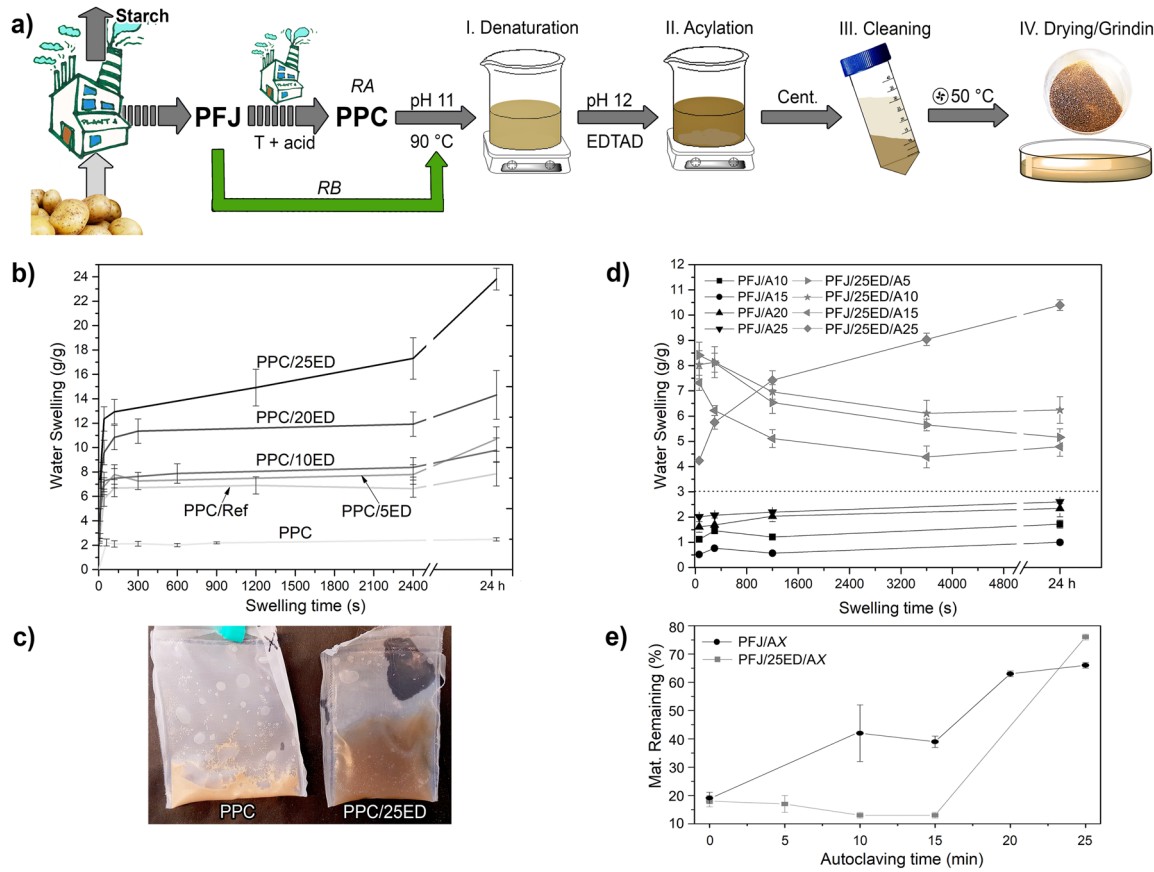

**Fig. 1 Functionalisation protocol of the potato protein and swelling results. a** Experimental protocol used for the functionalisation of the PPC (RA) and PFJ (RB). **b** Water free swelling capacity (FSC) of the acylated PPC samples at different EDTAD concentrations. **c** Representative sample of the swollen PPC and PPC/25ED (24 h). **d** Water FSC of the autoclaved PFJ and acylated PFJ samples. **e** acylated PFJ material remaining in the tea-bags after the 24 h water swelling at different autoclaving times. The dotted line in **d** separates the swelling transition between non and acylated PFJ samples. The "X" in **e** corresponds to the autoclaving time for the samples. Error bars represent ± one standard deviation.

develop sustainable SAPs with properties that resemble synthetic materials used in commercial applications where liquid retention under compressive force is required, e.g., in diapers.

To reduce the environmental impact of the extensive acidic/ heating treatments in producing industrial PPC, which result in extensive aggregation/cross-linking of the proteins, the acylation was directly performed on as-received PFJ (Fig. 1a, route RB). Autoclaving of the PFJ, independent of the time used, resulted in a protein that remained in the tea-bag (to a high extent) during the swelling test (Fig. 1d), which was not the case for the dry PFJ powder (PFJ/Ref). The drying of the tea-bags with the swollen PFJ material allowed the estimation of the amount of material remaining in the bag after the swelling test (24 h), as shown in Fig. 1e. The dry PFJ (not autoclaved, PFJ/Ref), resulted in only 18% of the material remaining in the bag as compared to PFJ material autoclaved for 10, 20, and 25 min and resulting in 40, 61, and 65% of the PFJ material remaining in the tea bags, respectively (Fig. 1e). These results also corresponded with the fact that PFJ had a clear liquid phase after 20- and 25-min autoclaving compared to the PFJ samples that were 10- and 15-min autoclaved (Supplementary Fig. 1). In general, an increase in autoclaving time resulted in an increase in the FSC of the PFJ samples (Fig. 1d), reaching the same swelling level as obtained for the as-received PPC, i.e. $2\,g\,g^{-1}$ (Fig. 1d).

The in situ EDTAD functionalisation of the PFJ and further autoclaving of the treated PFJ suspension showed a notable increase in the swelling of up to $9\,g\,g^{-1}$ within the first minute, followed by a progressive decrease in the FSC (Fig. 1d).

Furthermore, functionalisation of PFJ followed by autoclaving for 25 min (PFJ/25ED/A25) resulted in a typical swelling curve that resembled the behaviour of the functionalised PPC (Fig. 1b). The swollen gel-like material is shown in Supplementary Fig. 2. The PFJ/25ED/A25 had a swelling capacity of 9 and $10\,g\,g^{-1}$ after 30 min and 24 h, respectively (Fig. 1d). The drying of the tea bags containing the swollen samples autoclaved for 10 and 15 min showed that less than 20% of the initial material remained in the bag (Fig. 1e). Thus, the EDTAD treatment affected the solubility of the protein as the solely autoclaved PFJ had 45% material remaining (Fig. 1e). However, more than 75% of the material was retained within the tea bag after 25 min autoclaving of the in situ functionalised PFJ, which correlates with the progressive increase in the swelling observed for this sample with no evident swelling decrease (Fig. 1e).

The increase in the water FSC of the EDTAD-treated PFJ that was autoclaved for 25 min (PFJ/25ED/A25), was ca. 3 times higher than that of the PPC/Ref and the PFJ/A25 samples. This demonstrates the important contribution of the EDTAD, which acylates the lysine groups and other moieties in the potato protein, leaving pendant carboxylic acid groups[32,41,50,51]. The EDTAD acylation of the protein moieties have been reported to contribute to electrostatic repulsion and further swelling of the dry powders forming hydrogels[32,51]. For comparative purposes, a 25 min autoclaved PFJ sample was functionalised with 25 wt% EDTAD (PFJ/A25/25ED), resulting in a gel having a 24 h FSC of $6\,g\,g^{-1}$, which was lower than that for the PFJ/25ED/A25 ($10\,g\,g^{-1}$) (Supplementary Fig. 2). This result suggests that the autoclaving

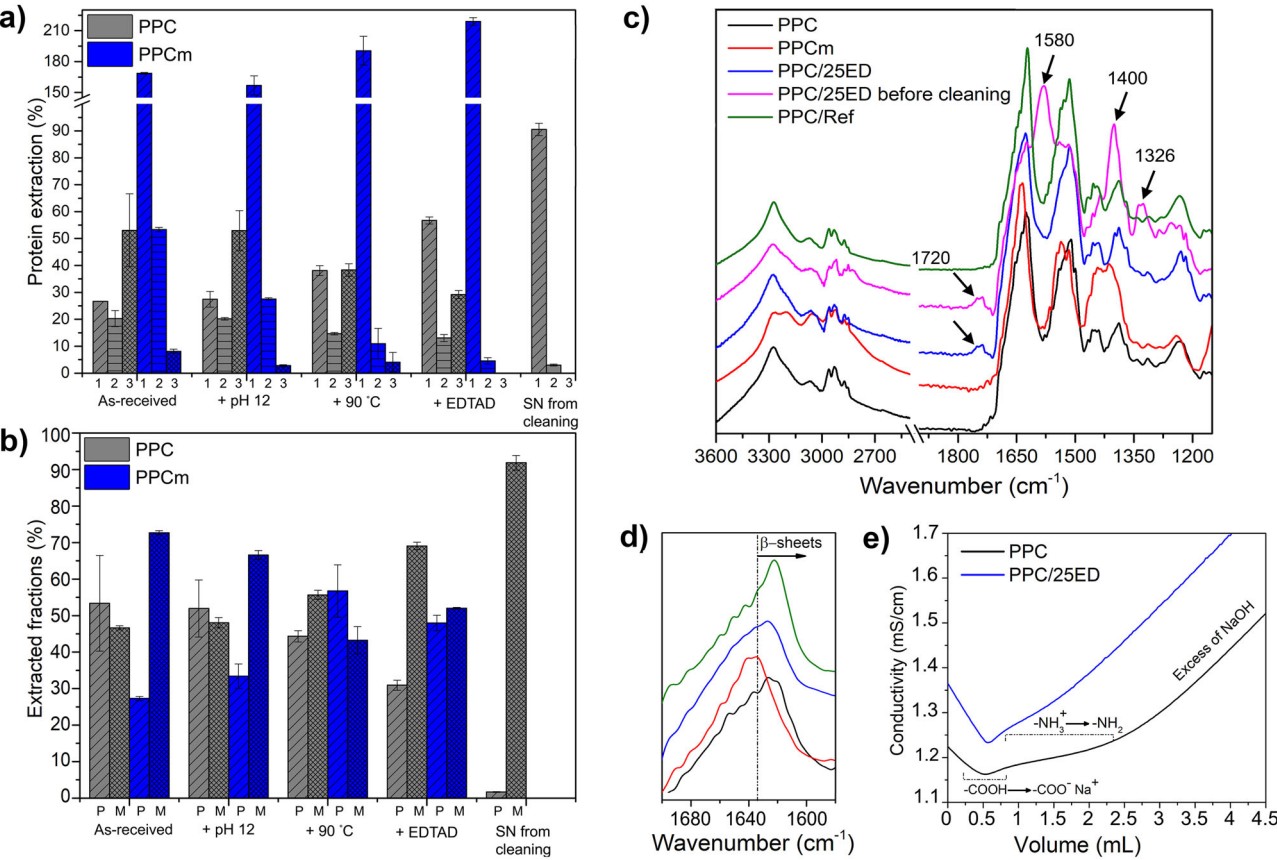

**Fig. 2 Protein extraction profiles and functionalisation. a** Three-step (1–3) protein extraction profiles, and **b** relation between polymeric (P) and monomeric (M) protein fraction (**b**) for conventional protein concentrate (PPC) and mildly extracted potato protein concentrate (PPCm) after the different functionalisation stages. The total extractable proteins for the as-received PPC were used as the normalisation value. SN from cleaning corresponds to a lyophilised aliquot of the SN of the cleaning of the PPC/25ED sample. Error bars represent ± one standard deviation. **c** Full FT-IR spectrum and **d** close-up of the amide I region. **e** Conductometric titration results.

before the EDTAD functionalisation reduces the available lysine and/or other amino acid moieties in the potato protein for reaction with the EDTAD, thereby reducing the efficiency of the acylation, as previously reported for wheat gluten proteins[41,43].

The increase in the swelling capacity of the EDTAD-treated PFJ, and the ability to form a protein network by the use of autoclaving (with no chemical cross-linking), pave the way for future optimisation work to obtain protein-based SAPs using the protein directly in PFJ, i.e., before the extensive industrial protein aggregation process used to obtain PPC.

**Protein functionalisation**. The higher degree of aggregation/polymerisation of the potato proteins in the PPC than in the PPCm (in-house mildly extracted PPC) was clearly depicted by Ext.1 and Ext.2 values above 100% in the PPCm (values were normalised to total extractable proteins from as-received PPC; Fig. 2a). The results correspond to previous studies showing that PPC was too cross-linked for the proteins to be extractable with the methodology applied[8,15,18]. Generally, the Ext. 1 values increased for both PPC and PPCm with the protein functionalisation stages evaluated (Fig. 2a). The as-received PPC had an initial protein extraction of ca. 25% (Ext. 1) and ca. 55% after the EDTAD acylation. The same trend was observed for PPCm, with initial protein extraction (Ext. 1) of ca. 165% and an extraction of ca. 210% for the acylated PPCm (Fig. 2a). The result indicates a gradual increase in cleavage of secondary protein bonds (SDS treatment), especially after the EDTAD acylation was performed, resulting in more easily extractable proteins. For PPC, extensive

sonication (Ext. 3) was required to extract the highly aggregated/cross-linked proteins of the industrial PPC. The proteins of PPCm were substantially more easily extractable, with the majority of the proteins extracted during Ext. 1 and Ext. 2 (Fig. 2a). In addition, almost no pellet was visually evident from the PPCm after Ext. 3, illustrating the high solubility of the aforementioned sample. Even after the EDTAD acylation, the PPC showed a lower protein extractability than the PPCm. However, the acylated PPC samples formed a cohesive network hydrogel contributing to a reduced leakage from the tea bag during testing and therefore resulting in a higher swelling compared to the as-received PPC (Fig. 1).

The differences in protein cross-linking between the PPC and the PPCm were also shown by the proportion of polymeric and monomeric proteins extracted during protein extraction (Fig. 2b). For example, a higher ratio of monomers than polymeric protein was extracted in PPCm as compared to PPC, suggesting that the mild protein extraction contributed to a less cross-linked protein network (Fig. 2b). The functionalisation of PPC resulted in an increase in the proportion of monomeric protein (Fig. 2b) extracted together with the total increase in protein extractability (Fig. 2a). These results suggest that a less cross-linked network was formed in the functionalised PPC as compared to the as-received PPC. Thus, for the highly cross-linked and aggregated industrial PPC, the functional EDTAD treatment contributed to a decrease/re-organisation of the protein cross-links, which might have contributed to an increase of the swelling capacity of the material. The PPCm showed an increase in the extracted

polymeric protein, from 25% (as-prepared) to 45% (EDTAD acylated), as shown in Fig. 2b. The increase in the polymeric fraction for the PPCm system, coupled with the increase in the PPCm solubility, indicate an efficient EDTAD acylation, which enhances the electrostatic repulsion between the protein chains, thereby increasing the hydrodynamic volume of the protein.

The HPLC profile of the lyophilised supernatant (SN) after the cleaning of the PPC/25ED (SN from cleaning), showed that it consists predominately of monomeric proteins, not incorporated into the cross-linked protein network (Fig. 2a and b). Besides, no pellet was evident after the three extractions from this sample. The result showed that the cleaning step, not only removed unreacted EDTAD from the system but also aids in the removal of soluble monomeric protein fractions. This is important from a commercial application point of view because such properties are needed for developing a safe superabsorbent with reduced leakage of material out of the product.

The higher total protein extractability of the EDTAD functionalised PFJ samples (>240%) compared to the as-received PFJ (105%), showed the PFJ solubility increased after the acylation treatment (Supplementary Fig. 3a). The acylated and autoclaved PFJ samples also resulted in a considerable increase in the extractable polymeric fractions (above 70%) compared to the non-autoclaved PFJ (25%), regardless of the autoclaving time used (Supplementary Fig. 3b). The acylated and 25 min autoclaved sample (PFJ/25ED/A25) had a lower protein extractability (170%) compared to the other functionalised and autoclaved samples (>240%), which corresponded to the good network cohesion of the sample, forming stable gels during the swelling tests (Fig. 1c and Supplementary Fig. 3a).

The FT-IR spectrum of the PPC/25ED sample demonstrated that the cleaning process was effective in removing free EDTAD (Fig. 2c). The peaks for the ring-opened EDTAD (free EDTAD) are located at 1580 and 1400 cm$^{-1}$ and are assigned to the stretching (asymmetric and symmetric, respectively) of the C=O from the carboxylate (–COO$^-$), and the 1326 cm$^{-1}$ band is assigned to the stretching (C–O) of the tetrasodium EDTA (–CH$_2$(COO)$^-$), which is the main product in the free EDTAD[34,52,53]. The band at 1720 cm$^{-1}$ in PPC/25ED before cleaning, assigned to the stretching of the C=O in the –COOH group[52–55], was also present in PPC/25ED and absent in PPC/Ref. The peaks at 1580 and 1400 cm$^{-1}$ were also more intense in PPC/25ED compared to PPC/Ref, suggesting that the EDTAD acylation of the PPC increased the carboxylic acid/carboxylate content of the protein. The amide I region of PPC was shifted to a lower wavelength compared to PPCm, indicating the presence of a dominant aggregated β-sheet protein structure in PPC (Fig. 2d)[17]. The PPCm sample showed sharp peaks at 1550 and 1100 cm$^{-1}$ that were ascribed to ammonium sulphate that was not fully washed from the PPCm during the dialysis[56].

The curve resolution of PPC indicated that 50% of the protein structure is strongly bonded β-sheets, whereas only 29% was assigned to that protein structure for PPCm (Table 1)[57]. The secondary structure agrees with the SE-HPLC results, showing the

highly aggregated/cross-linked nature of the industrial PPC. The PPC/25ED sample showed a reduction in the strongly bonded β-sheets compared to PPC (Table 1), suggesting that EDTAD reduces the aggregation level of the protein. The highly aggregated structure of the PPC is, however, a determinative factor in the ability to produce an acylated PPC that has a stable network by the endogenous cross-links and structure of the protein (Fig. 1b), therefore not leaking from the bag during the FSC test (Fig. 1).

The FT-IR on the lyophilised PFJ showed intense peaks at 1054 and 1390 cm$^{-1}$, assigned to C–O stretching and C–H symmetric bending of potato starch (remaining in the PFJ after the industrial separation; Supplementary Figure 4a), respectively[58,59]. The peak at 1568 cm$^{-1}$ was assigned to N–H in-plane bending vibration from potato protein isolate (Supplementary Fig. 4a)[60]. The remarkable decrease of the aforementioned peaks in PFJ/A20 and PFJ/A25 (Supplementary Fig. 4a) suggests reactions involving N–H in the potato protein and cleavage of C–O in the potato starch, and also corresponds with the FSC of these samples not leaking from the tea bags (Fig. 1). The FT-IR revealed no apparent differences in the acylated PFJ samples, but a shift of amide I region towards more β-sheets was observed with an increase in the autoclaving time (Supplementary Fig. 4b). Similarly, no differences in the FT-IR were observed between PFJ/A25, PFJ/A25/25ED, and PFJ/25ED/A25, although the later sample had more β-sheets and stronger C–O signal from potato starch (Supplementary Fig. 4c).

The conductometric profile of the PPC/25ED indicated that the region ascribed to the lysine deprotonation has been shortened compared to the untreated PPC (Fig. 2e). The result suggests that the lysine content (pendant amino groups in the protein) has been reduced by the condensation of the EDTAD on the lysine during the acylation reaction, as suggested in a previous work[61]. The current results, showing opportunities to produce a bio-based swellable product through EDTAD acylation of a highly insoluble protein, opens-up possibilities for implementing the EDTAD reaction on more insoluble protein sources while still using aqueous systems.

**Sample structure and biodegradability.** The PPC/25ED particles generally had a flat appearance with an average length of 304 ± 135 µm. The high magnification images of the surface showed a relatively smooth and pore-free feature (Fig. 3a, b). The particle size of the commercial SAP product was 406 ± 167 µm, and the surface of the PPC/25ED resembled that of the SAP particles (Fig. 3e-f). However, the appearance of the PPC/25ED material after 24 h MQw swelling and subsequent lyophilisation resembled instead the structure of a fragile fragmented foam (Fig. 3c). The average size of the foam-like material was 552 ± 269 µm, with interconnected pores and cell walls (with a size of ca. 20 µm), as shown in Fig. 3d. The increase in the particle size and a structure resembling a foam illustrate a large amount of water present in the swollen PPC/25ED that was initially frozen and then sublimated during the lyophilisation procedure[50].

| Peak (cm$^{-1}$) | Assignment | Samples (%) | | |
|---|---|---|---|---|
| | | **PPCm** | **PPC** | **PPC/25ED** |
| 1618, 1625 | β-sheets, strongly bonded | 29 | 50 | 41 |
| 1634, 1680 | β-sheets, weakly bonded | 18 | 5 | 3 |
| 1644, 1651, 1658 | α-helixes and random coil | 28 | 22 | 29 |
| 1667, 1691 | β turns | 25 | 23 | 27 |

**Table 1 Relative content of protein secondary structure (%) from the FT-IR curve resolution.**

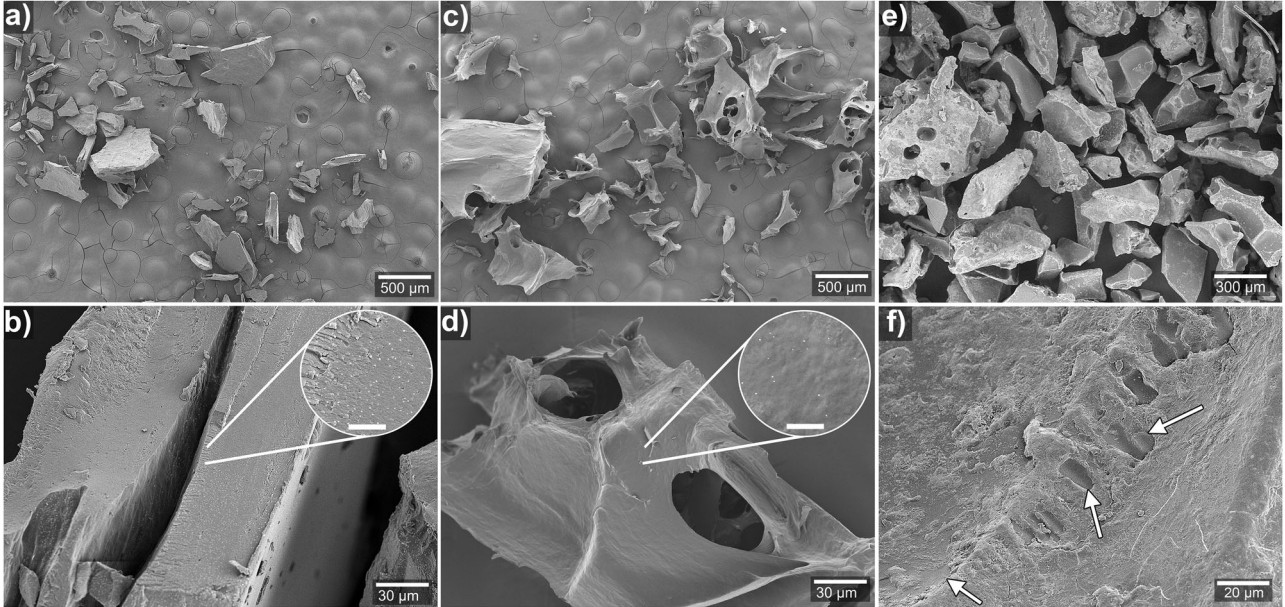

**Fig. 3 Particle morphologies. a, b,** Dry PPC/25ED particles. **c, d,** Lyophilised PPC/25ED foam-like particles after 24 h in MQw. The scale bar in the inset of **b** and **d** is 2.5 μm. **e, f,** Images of commercial SAP particles. The arrows in **f** point at the smooth-surface regions on the surface of the particles.

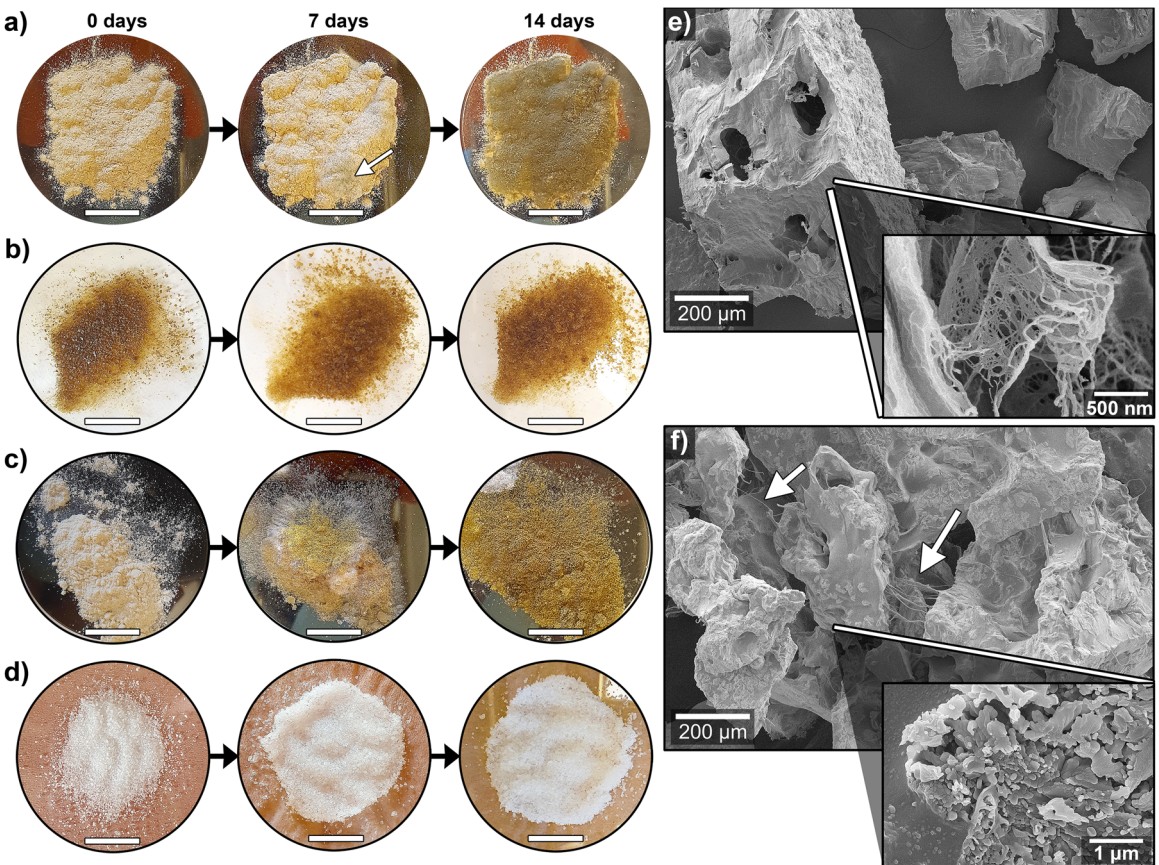

**Fig. 4 Mould growing and sample morphology.** Samples exposed to 100% RH: **a,** PPC, **b** PPC/25ED, **c** WG, **d** SAP. The white arrow in **a** point mould growing sites. **e** Images of PPC/25ED, and **f** commercial SAP particles after being exposed two weeks to 100% RH and then lyophilised. The white arrows in **f** point at some of the observed filaments connecting the SAP particles. The scale bar in the photographs represents 1 cm.

The PPC and wheat gluten (WG) reference material showed rapid mould growth on the powder after 2 weeks of exposure to 100% RH, as shown in Fig. 4a, c, respectively. In contrast, the PPC/25ED visually resembled the synthetic SAP material, i.e., without any apparent mould growth on the powder after 2 weeks exposure to 100% RH, Fig. 4b, d, respectively. This property was also obtained for PPC/Ref, which resisted mould growth even after 2 weeks of being in contact with the as-received PPC that

was completely covered in mould (Supplementary Fig. 5). The result agrees with previous work on chitosan, showing that the addition of charged groups onto polymer substrates could inhibit the protein adsorption and cell adhesion necessary for bacterial/mould growth[62–65]. An increase in the degree of acetylation of polymers has resulted in even more significant bacterial growth inhibition, which has been ascribed to the added functional groups on the polymers disrupting the regular mass transport through the cell walls of the bacteria[66]. The mould growth process could be then influenced by the alkaline and EDTAD functionalisation implemented on the PPC (inducing the formation of the charged groups), thereby inhibiting its regular proliferation (Fig. 4a). The results indicate that the acylation process on potato protein effectively yields a material with adequate storage properties even at high relative humidity conditions. The PPC/25ED material also showed a moisture uptake of 0.9 and 1.1 g g$^{-1}$ after 1 and 2 weeks of exposure to 100% RH, respectively. This result demonstrates the ability of the acylated PPC material to take up water from the saturated atmosphere in a similar way as the tested synthetic/commercial SAP, which had an uptake of 2.6 and 3.7 g g$^{-1}$ after 1 and 2 weeks, respectively. The aforementioned property coupled with the absence of extensive mould growth on the PPC/25ED shows the possibility to use the material, after further optimisation of the material's uptake at different RHs, as a disposable desiccant.

The SEM images of the lyophilised PPC/25ED after 2 weeks of 100% RH exposure reveal that the particles had increased in size to $412 \pm 107 \, \mu m$, and pores of $91 \pm 33 \, \mu m$ in diameter were observed, which resulted from the sublimated water after the lyophilisation process (Fig. 4e). The high magnification of the PPC/25ED particles (Fig. 4e) revealed a fibrillar structure on the surface, which resembles aspects of mould[67,68]. The morphology of the lyophilised SAP suggests that the particles have merged into clusters sized above 500 μm, connected by 5–10 μm thick filaments (Fig. 4f). At high magnification, the SAP showed regions with porous structures, suggesting an irregular moisture uptake in the sample and indicated areas of pre-existing phases in the dry SAP particles (Supplementary Fig. 5b). The different phases may have originated from polyacrylic acid with varying degrees of cross-linking that after moisture uptake and lyophilisation resulted in localised porous SAP structures (Fig. 4f).

Figure 5 shows the rapid biodegradation of the PPC/25ED, measured by the emitted $CO_2$ obtained from the activity of the microorganisms present in the soil. In fact, ca. 30% of the PPC/25ED was degraded 10 days after the start of the test. The sample showed a ca. 50% degradation after 97 days, which was in the same range as the control sample (potato starch) and previously reported $CO_2$ emissions from bio-based materials[5]. The synthetic SAP showed, however, no signs of biodegradation after 97 days, and gel particulates were visually observed randomly distributed in the soil (Fig. 5, inset). These results demonstrate the high biodegradability of the bio-based PPC absorbent material in comparison to synthetic SAP. The biodegradation of the PPC/25ED is an advantage for the design of future products requiring SAP technology, especially in items that are disposable and of single-use, e.g. diapers and sanitary pads, thereby providing a material that can be biodegraded safely in the environment. The result paves an avenue for future work related to the in-depth understanding of these protein-based superabsorbents' degradation mechanism and the specific microorganism action on the material.

**GHG assessment**. The total energy involved in the PPC acylation and the in situ PFJ acylation was assumed to be equal for process schemes A and B (Fig. 1a). Therefore, only the energy

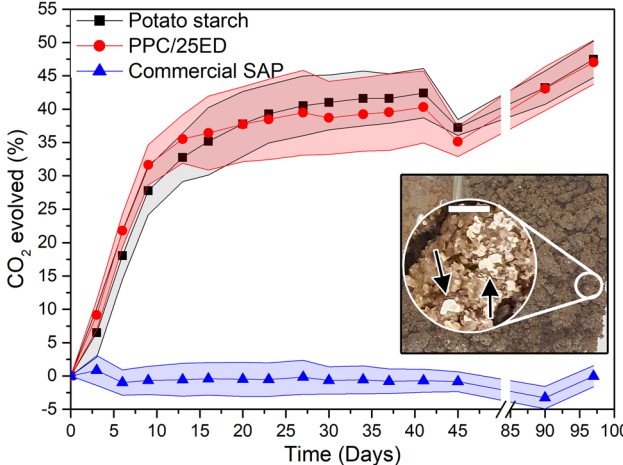

**Fig. 5 CO$_2$ evolved from the samples during the soil degradation test.** The inset shows the top of a container with soil and swollen SAP particles after 97 days of testing. The arrows in the inset point at commercial SAP gel particles in the soil after 97 days from the start of the test. The scale bar in the inset represents 2 mm. The coloured areas represent the standard deviation range of the average (solid individual data point).

required to produce the industrial PPC was estimated, as this represents the additional energy needed if the acylation will be performed on the PPC instead of the PFJ. The amount of natural gas required to heat the PFJ (to coagulate the protein) and to evaporate the water from the protein precipitate was 9.0 and 3.1 kg per cubic metre of PFJ, which resulted in the emission of 24.3 and 8.4 kg of fossil $CO_2$, respectively. At an approximate annual global production of 3.7 million tons of potato starch, resulting in ca. 10 million m$^3$ of PFJ, the calculated annual GHG savings of performing the in situ acylation of PFJ instead of the PPC for producing protein-based SAPs is ca. 325 000 tons of $CO_2$. Assuming that a passenger vehicle consumes 5 L of fuel per 100 km, travels 15 000 km/year, and has fuel with a carbon concentration of 3.1 kg $CO_2$-eq. per litre, the GHG savings of the in situ PFJ corresponds to the amount of GHG that is emitted from ca. 116 000 passenger vehicles annually.

The GHG savings corresponded to ~10% of the life-cycle GHG emissions of ca. 3.2 million tons $CO_2$-eq annually originates from the amount of starch produced in upstream processes (using an emission factor of 877 kg $CO_2$-eq per ton starch)[23].

At a final product output of 14.2 kg of SAP per cubic metre of PFJ, this corresponds to a GHG savings of 2.3 kg $CO_2$ per kg of protein-based SAP produced. Energy and GHG emission savings could be 5–10% higher if the drying process would employ combined heat and power production or heat recovery technology, e.g., for pre-warming the PFJ[69].

**Conclusion**

The acylation of highly aggregated and low soluble industrial side-stream PPC using EDTAD resulted in an increase of the swelling capacity of the PPC from 2 to 24 g g$^{-1}$ (2400%). Although the protein aggregation in the PPC decreased the acylation efficiency, the endogenous cross-links of the industrially processed protein yielded a stable hydrogel without the need for toxic cross-linkers, which represents an advantage compared to other proposed protein-based SAPs. Here, the acylation of the soluble potato protein contained in the PFJ, followed by autoclaving to aggregate/cross-link the protein, resulted in a protein network with a swelling capacity of up to 11 g g$^{-1}$ in water without a large material loss. This result opens-up opportunity for future optimisation work in the acylation of the PFJ before the

severe industrial treatment to produce PPC, which could result in greenhouse emission savings compared to the acylation process using PPC. The high blood and water swelling combined with the moisture uptake of the acylated PPC make the material a promising candidate for applications where today synthetic SAP are the material of choice, e.g., in sanitary pads and desiccants. The acylated PPC showed rapid biodegradation, i.e., 50% degradation after 97 days, while the commercial SAP did not show any signs of degradation during the same time. The possibilities of producing a protein-based absorbent material with tuneable swelling capacity are foreseen as an important contribution to the UN Sustainable Development Goals, as it allows for more sustainable production of bio-SAP and the utilisation of a disposable material that can degrade into safe components permitting a responsible product consumption while simultaneously contributing to material circularity. Future work should focus on the industrial scalability of the method, especially in the area of unreacted EDTAD recovery and material performance over time in storage.

## Methods

**Materials.** Commercial PPC and PFJ were provided by Lyckeby Starch AB, Sweden. The PPC protein content was 82 ± 2%, and the protein content was 31 ± 7% of the solids content in the PFJ (determined as described in Protein content analysis). The moisture content of PPC was 8.1 ± 0.4%. EDTAD (98%), sodium hydroxide (NaOH, ≥99%), sodium bisulphite (NaHSO₃, ACS Reagent), and ammonium sulphate ((NH₄)₂SO₄, >99%) were purchased from Sigma-Aldrich, Sweden.

To evaluate possible effects of the acylation protocol on the PPC, mildly extracted potato protein was produced in-house using ammonium sulphate precipitation[70–72]. Approximately 1 kg of potato tubers were thoroughly washed with water, peeled and rinsed with 2 wt% NaHSO₃ solution to avoid browning. The peeled and washed potatoes were juiced in an Angelia 5500 twin-screw press (Angel Co. ltd., South Korea), and the pressed potato juice was left for 15 min to settle and thereafter centrifuged at 3000 × g (15 min) at 10 °C. The SN was vacuum filtered (30–50 μm filter paper), and to ensure the quality of the potato juice the pH was measured using a pH-metre (Mettler-Toledo, Switzerland), which indicated a pH of 6.9. Then, (NH₄)₂SO₄ was added to the juice up to 60% of the reported saturation limit (542 g/L at 25 °C). The pH of the 60%-saturated solution was corrected to 5.7 by dropwise addition of 0.5 M H₂SO₄. The solution was left under constant stirring at 4 °C for 1.5 h to precipitate the potato proteins, and the suspension was then centrifuged at 3000 × g for 30 min (at 4 °C). To remove the excess salt, the protein pellet was re-suspended in Millipore water (MQw) and dialysed extensively against MQw with a dialysis membrane with a 10–30 kDa cut-off. The dialysed mild extracted potato protein (PPCm) solution was frozen at −80 °C and lyophilised for 72 h and ground using a mortar and pestle, yielding a beige-coloured product. The protein content of the PPCm was 94 ± 0.5%.

**Protein content analysis.** The nitrogen content of the materials was determined using the Dumas method with a Flash 2000 N/C Analyser (Thermo Scientific, USA). Briefly, ca. 3 mg of the materials were placed into thin tin capsules, which were folded, and pressed to remove the excess air. The nitrogen-to-protein conversion factor used was $N \times 6.25$[8], and the results were presented as averages from triplicates.

**Acylation of potato protein concentrate.** The as-received PPC powder was acylated as previously reported by Capezza et al[41]. and Cuadri et al[45]. for wheat gluten and soy protein, respectively, with a few modifications[41,45]. The PPC powder was added to a beaker containing MQw pre-adjusted to pH 11 using 1 M NaOH (Fig. 1), forming a 2 wt% protein suspension. The addition of the PPC took ca. 10 min with the pH continuously re-adjusted to pH 11. To enable protein denaturation, i.e., unfolding and exposing protein functional groups, the solution was heated using a water bath (preheated to 70 °C) to 90 °C for 30 min (Fig. 1, step I). The solution was then rapidly cooled to room temperature using an ice bath. For the acylation (Fig. 1, step II), the pH of the suspension was adjusted to pH 12 and EDTAD was gradually added to the suspension. The total time taken to add the EDTAD into the PPC suspension was 30 min, and the pH was corrected to pH 12 when needed. Different amounts of EDTAD were used; 5, 10, 20, 25 wt% (w/w) with respect to the PPC content and 1:1 EDTAD:protein mass ratio. After finalising the addition of EDTAD, the suspension was stirred for 1.5 h, while continually monitoring the pH to keep it at pH 12 using a pH-metre. For removing the unreacted EDTAD salts (Fig. 1, stage III), the suspension was centrifuged at 4200 rpm (3000 RCF) for 10 min using a Sorvall RC6 + Centrifuge (Thermo Scientific, USA). The SN was decanted and replaced by fresh MQw pre-adjusted to pH 11, the pellet was re-dispersed and vortexed for 10 min, and again centrifuged as previously described. Thereafter, the SN was decanted and fresh MQw was added up to 25% of the initial

volume. The pellet was vortexed for 10 min, and the pH of the suspension was neutralised to pH 7–8 using 1 M HCl. The thick suspension was poured into a glass petri dish and dried overnight in a forced-air oven at 50 °C (Fig. 1, stage IV), and then ground to particles using a mortar and pestle. An identically processed PPC sample was produced as a reference sample, i.e., without any addition of EDTAD (PPC/Ref). All powders were stored in a desiccator with silica gel for at least 1 week before test/analysis. The samples with EDTAD were labelled PPC/XED, where X denotes the EDTAD concentration used.

**Acylation of potato fruit juice.** As-received PFJ was denatured and acylated as described above, with some modifications. For the acylation reactions, the total protein content of the PFJ was assumed to 2 wt%. Due to the high solubility of the acylated potato proteins in the PFJ, they were precipitated by adjusting the pH of the solution to 3–3.5 using 1 M HCL at room temperature. The suspension was then centrifuged at 4200 rpm (3000 RCF) for 10 min and the SN was decanted and replaced with fresh MQw pre-adjusted to pH 3.5. The pellet was re-suspended using a spatula, vortexed for 10 s, shaken at 250 rpm for 10 min, and then centrifuged as previously described. The SN was poured out and individually replaced with MQw (only 25% of the initial total volume). The pellet was re-dispersed, vortexed, shaken, and the pH adjusted to pH 7–7.5 using 1 M NaOH. Thereafter, the acylated proteins from the PFJ were separately autoclaved for 5, 10, 15, and 25 min at 120 °C. The thick suspension of acylated and autoclaved proteins was placed into a petri dish, forced-air dried at 50 °C overnight, and ground to particles using a mortar and pestle. The samples were labelled as PFJ/25ED/AX, where "X" denotes the autoclaving time used. A non-autoclaved EDTAD-acylated PFJ sample (PFJ/25ED) and a dried fraction of the as-received PFJ (non-autoclaved and non-acylated, PFJ/Ref) were also prepared. To evaluate the possible effects of the autoclaving process in the acylation of the protein from PFJ, the PFJ was also autoclaved before the EDTAD-acylation. This sample was acylated and cleaned as described previously. The images of the PFJ before and after the autoclaving process is shown in Supplementary Figure 1.

**Liquid swelling and retention.** The liquid FSC of the samples was determined following the Nonwovens Standard Procedure (NWSP) 240.0.R2, referred to as the "tea-bag test"[73]. A monofilament fabric (25–50 μm mesh size, equivalent to 442–305 mesh) purchased from Sintab Produkt AB, Sweden, was used for producing the tea bags. The bag dimensions were ca. 40 × 60 mm, three edges were heat-sealed, and the weight of each bag was measured ($W_b$). To perform the test, 100–200 mg of the dry powders ($W_d$) were put inside the bags, the fourth side was then sealed, and the filled bags were kept in a desiccator for a minimum of 6 h before testing. The bags were hooked to a rod and individually immersed in beakers containing an excess amount of MQw, saline (0.9 wt% NaCl in water), or defibrinated sheep blood, respectively. The immersion times chosen were 1, 5, 10, 30 min, and 24 h. After the corresponding immersion time, the bags were removed from the liquid, hung for ca. 30 s, allowing the drainage of the excess water. Thereafter, they were gently placed on tissue paper for 10 s removing surface water. The weight of the bags was then measured ($W_i$) in triplicates, and the average and standard deviation were reported. For the blank, three empty dry bags ($W_o$) were handled identically as before, and the blank bags were allowed to soak in MQw, saline or blood for 24 h. The wet weight of the blank bags ($W_w$) was used to obtain a correction factor ($W'$). An additional FSC test was performed using saline (0.9 wt% NaCl in deionised water) with a measured conductivity of 16 mS/cm (Saline A). The electric conductivity was measured using a Seven Compact S230-Basic Conductivity benchtop metre (Mettler-Toledo, USA). Saline A was used as it is a standard liquid used in the sanitary industry to assess the quality of synthetic SAPs. The centrifugal retention capacity (CRC) was estimated by centrifuging the tea-bags with the swollen materials (after 30 min swelling time) at 1230 rpm (250 RCF) for 3 min, and then weighing the samples ($W_c$). The FSC was calculated according to Eqs. 1 and 2 and the CRC using Eq. 3.

$$W' = W_w/W_o \tag{1}$$

$$\mathrm{FSC}\left(\frac{g}{g}\right) = \left[W_i - (W_b \times W') - W_d\right]/W_d \tag{2}$$

$$\mathrm{CRC}\left(\frac{g}{g}\right) = \left[W_c - (W_b \times W') - W_d\right]/W_d \tag{3}$$

**Size-exclusion high-performance liquid chromatography (SE-HPLC).** A Waters 2690 HPLC (Waters, USA), equipped with a Phenomenex BIOSEP SEC-4000 column (Phenomenex, Denmark) and a Security Guard GFC 4000 pre-filter (Phenomenex, Denmark) was used. The proteins were extracted using three extraction steps, as previously reported by Gällstedt et al[74]. For the extractions, a buffer solution consisting of 0.5 wt% sodium dodecyl sulphate (SDS) and 0.05 M NaH₂PO₄ (pH 6.9) was used. A total of 16.5 ± 0.05 mg of the protein powders were dispersed in 1.4 mL of the SDS-phosphate buffer. The suspension was vortexed for 10 s, followed by shaking at 2000 rpm for 5 min. The suspension was then centrifuged at 12 500 rpm (9000 RCF) for 30 min using a Legend Micro 17 Sorvall Centrifuge (Thermo Scientific, Sweden). The extractable soluble proteins from the

SN were decanted and defined as Extraction 1 (Ext. 1). For the second extraction (Ext. 2), the pellet from Ext. 1 was re-dispersed in 1.4 mL of fresh buffer, ultra-sonicated using a Soniprep 150 Ultrasonic Disintegrator (Sanyo, UK) at an amplitude of 5 μm for 30 s, and centrifuged as described previously. The SN was decanted and the soluble proteins were assigned to Ext. 2. For the third and final extraction (Ext. 3), the pellet was re-dispersed in 1.2 mL of the SDS-phosphate buffer, sonicated for 30 + 60 + 60 s, and centrifuged as previously described. Triplicates were used for each sample, and the total protein extractability was normalised against the total extraction of commercial PPC. The injections consisted of 20 μL of the SN, using a 0.2 mL/min isocratic 1:1 flow of acetonitrile (0.1% TFA) and MQw (0.1% TFA). The chromatograms were obtained using a Waters 996 Photodiode Array Detector (Waters, USA) at a wavelength of 210 nm, and the extraction profile was divided as polymeric (from 1 to 15 min elution time) and monomeric (15 to 26 min elution time) protein fractions.

**Fourier-transform infrared (FT-IR) spectroscopy**. The FT-IR spectra were obtained using a PerkinElmer Spectrum 100, an ATR single-reflection Golden Gate unit (Graseby Specac LTD, England), and a triglycine sulfate detector. The scanning resolution was 4.0 cm$^{-1}$ and the spectrum was obtained based on 32 consecutive scans. The peak deconvolution of the protein amide I region (1700–1580 cm$^{-1}$) was performed as previously described by Cho et al[2]. using the PerkinElmer Spectrum software (version 10.5.1), setting the enhancement factor (γ) and a smoothing filter to 2% and 70%, respectively. The individual protein secondary structure fractions were obtained by a Gaussian curve resolution, fitting the amide I region to 9 peaks; 1618, 1625, 1634, 1644, 1651, 1658, 1667, 1680, and 1691 cm$^{-1}$. The peak centres were allowed to move ±1 cm$^{-1}$ if necessary for improving the fitting result.

**Sample morphology**. The acylated protein particle morphology was examined in a field-emission scanning electron microscope, FE-SEM S-4800 (Hitachi, Japan) and 3 kV was used during all the characterisation. The particles were placed onto conductive carbon tape and thereafter sputtered with a palladium/platinum (Pt/Pd) using a 208RH High-Resolution Sputter Coater (Agar, UK). The sputtering time was 45 s, which left a coating layer of ca. 4 nm. To study the morphological aspects of the swollen particles, the entire tea bag containing the material was frozen at −80 °C and thereafter lyophilised for 72 h. For the analysis, the lyophilised material was cryo-fractured using liquid nitrogen and prepared as previously described. The particle size was calculated as the average of 50 measured particles using the software ImageJ®.

**Particle charge density**. The charge density of the materials was obtained using a 702SM Titrino conductometric titrator unit (Metrohm, Switzerland). The SCAN-CM 65:02 procedure was used for the charge density determination, as described by Hollertz et al[49]. Briefly, 0.8 g of the protein powder was dispersed in 50 mL of MQw, the pH adjusted to pH 2 (1 M HCl) to ensure the proteins are fully protonated and left stirring for no longer than 25 min to prevent extensive protein hydrolysis. The suspension was then filtered and thoroughly washed with MQw removing the excess ions. The rinsing continued until the conductivity of the washing water was below 0.5 μS. For the conductometric test, 0.2 g of the protonated material was dispersed in a beaker containing 10 mL of 0.1 M NaCl, 5 mL of 0.01 M HCl, and 485 mL of MQw. The conductometric titration was performed in duplicates, using 0.1 M NaOH and a continuous flow of nitrogen gas.

**Biodegradability and mould resistance test**. The biodegradability of a commercial synthetic superabsorbent (SAP) and an acylated PPC was assessed by soil degradation as previously described by Muneer et al[5], following the standard ASTM D5988-03. The soil was obtained from the field at SLU, Alnarp (N55.661303, E13.077222), sieved to a particle size of 2 mm and stored at 5 °C. The initial pH of the soil was measured using a SevenCompact Duo S213 pH/conductivity-metre (Mettler-Toledo, Switzerland) by dispersing 2 g of the soil in 10 mL MQw. To determine the moisture content of the soil, 11 g of the soil was placed in an oven at 105 °C for 24 h, and the weight difference measured yielded the moisture content. The carbon and nitrogen ratio of all the materials tested were obtained using a Flash 2000 Analyser (Thermo-Fisher Scientific, USA).

For the degradation test, ca. 200 g of soil was placed in an air-tight plastic container of 1.8 L capacity (Supplementary Fig. 6a). To meet the ASTM standard, the pH of the soil was adjusted to pH 6 by adding 10 mL of 0.5 M ammonium dihydrogen phosphate ($NH_3H_2PO_3$), which was also used as a nitrogen source. A beaker containing 50 mL of MQw and a beaker containing 20 mL 0.5 M potassium hydroxide (KOH) were also placed in the plastic container (Supplementary Fig. 6a). The MQw was used for maintaining the atmospheric moisture content inside the container and the KOH for reacting with the carbon dioxide ($CO_2$) emitted from the biodegradation of the material. Approximately 1 g of the potato protein powders were homogeneously distributed in the soil. In separated plastic containers, 1 g of potato starch was distributed in the soil as a positive control, whereas another container was left with no sample (soil blank). Triplicates were prepared for all samples and the results are presented as the average. The containers were hermetically sealed and stored in a dark room at RT during the test. To determine the evolved $CO_2$ at a different intermediate time, the containers were opened and the beaker with 0.5 M KOH was titrated using 0.25 M HCl, using phenolphthalein as an indicator. The remaining KOH was stoichiometrically calculated according to Eq. 4, and then the absorbed $CO_2$ was calculated according to Eq. 5.

$$KOH + HCl \rightarrow KCl + H_2O \tag{4}$$

$$2KOH + CO_2 \rightarrow K_2CO_3 + H_2O \tag{5}$$

Once the titration was performed, fresh MQw (50 mL) and 0.5 M KOH (20 mL) were added to the beakers, and the container was sealed again. The emitted $CO_2$ was calculated by titration measurements every 3–4 days, and the $CO_2$ emitted by sample biodegradation was calculated, taking into consideration the $CO_2$ emissions solely by the soil (obtained from the soil blank). The moisture content and pH of the soil after the test were determined in the same way as previously described, using ca. 2 g of the soil.

Mould growth and humidity uptake tests were performed by placing ca. 0.5 g of a commercial SAP, PPC, PPC/25ED, and as-received wheat gluten protein WG (used as reference protein material), in an air-tight plastic container, with MQw in the bottom, as shown in Supplementary Fig. 6b.

**GHG assessment**. Direct acylation of the potato protein in the PFJ will avoid the need to produce PPC (from PFJ) as the raw material for the development of bio-based superabsorbents. Greenhouse gas (GHG) emission savings were accordingly assumed to correspond to the GHG emission from the industrial process for obtaining PPC from PFJ. This process was assessed as equivalent to the following steps: (i) heating of the PFJ stream for protein coagulation and (ii) evaporation of the water from the protein precipitate[20,26]. For the first step, the PFJ is heated from 22 to 120 °C under pressure so that no boiling occurs. The heat recovery from the process then heats up the PFJ from 22 to 65 °C, and the remaining temperature increase is performed using steam injection. The coagulated and precipitated PPC of the second step was assumed to have a water content of 60%. This water is completely evaporated from the protein precipitate to obtain the dry PPC. No heat recovery from the drying step was assumed. The acidic treatment to adjust the pH and separation processes of the PFJ was neglected, as they are considered to be less energy intensive. Natural gas was used in the calculation as an energy source to produce the saturated steam at 160 °C. The steam mass ($m_s$) needed was calculated according to Eq. 6, where $c$ is the specific heat (water: $w$ 4181 J kg$^{-1}$ K$^{-1}$; steam: $s$ 2080 J kg$^{-1}$ K$^{-1}$) and $L$ is the latent heat (water: $w$ 2265 kJ kg$^{-1}$), which was only considered in the evaporation step.

$$m_s = \frac{c_w \times m_w \left(T_f - T_{w,o}\right) + L_w \times m_w}{c_s \left(T_{s,o} - T_f\right)} \tag{6}$$

For the calculations, a steam generation energy efficiency of 85% was assumed[75]. Drying efficiency ranges between 20 and 90% have been previously reported[76], thus a 60% value was selected for use here. Typical natural gas composition was assumed, which is as follows: methane 96%, ethane 2%, propane 0.6%, and other alkenes 0.3%[77], resulting in a heating value of 49.4 MJ kg$^{-1}$ and carbon content of 74.2%.

## Data availability

The Supplementary Information includes the morphology of the PFJ before and after the autoclaving (Supplementary Fig. 1); the PFJ products after 60 min of water swelling (Supplementary Fig. 2); the three-step and total protein extraction profiles by HPLC (Supplementary Fig. 3); the FT-IR of the lyophilised autoclaved PFJ samples (Supplementary Fig. 4); mould growing test of the PPC/Ref material (Supplementary Fig. 5); and biodegradability test box (Supplementary Fig. 5). Supplementary Movie 1 is available online. The data that support the findings of this study are available from the corresponding authors upon reasonable request.

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

## Acknowledgements

This project was funded by VINNOVA (Grant 2015-03506) with the collaboration of Lantmännen ek för and Essity AB, Sweden. The environmental study was supported by South-Baltic Area Interreg project Bioeconomy in the south-Baltic area: Biomass-based Innovation and Green Growth (BioBIGG). Maria Luisa Prieto-Linde and Anders Ekholm are acknowledged for help with the SE-HPLC/CN content measurements. Dr. Annelie Moldin is acknowledged for the feedback regarding the raw materials provided within the project. The SLU Disease Vector Group and Dr. Sharon Hill are acknowledged for providing the defibrinated sheep blood used for the blood swelling tests. David Johnson, Mariano Parra, Mercedes Bettelli, and Wolfgang Suárez are acknowledged for their assistance during the biodegradation tests.

## Author contributions

A.J.C. performed the PPC experiments and designed the entire study. F.M. performed the PFJ experiments. T.P. conducted the environmental assessment. M.L. assisted with the swelling measurements. W.R.N., O.D., R.T.O., M.S.H., E.J. supervised the experiments, contributed to the data analysis, and assisted in the writing. All the authors discussed the results and commented on the manuscript.

## Funding

## Competing interests

The authors declare no competing interests.
