## [Peer Review File · Communications Chemistry]

Reviewers' comments:

Reviewer #1 (Remarks to the Author):

This paper presents interesting and novel results about the functionalization of potato protein obtained from agricultural side-streams. Thus, a superabsorbent material has been developed in a sustainable way, presenting a high water uptake capacity and biodegradability. So, I recommend its publication after the revision of the follow minor comments:

- 1) The definition of the principal objective and the characterization carried out should be improved in the introduction section. Now, they are difuse. In addition, the results should be ommited in this section.
- 2) Is there any hypothesis about why adding 15% EDTA worsens water swelling?
- 3) Line 6: Add the term "respectively" after blood.
- 4)Line 18: Change "Counterparts" for "materials".
- 5) Line 23: Change % for wt%.
- 6) The microorganisms present in the soil are a key parameter for the degradation of bioplastics. It has been taken in consideration?
- 7) Have analytical studies been carried out?

Reviewer #2 (Remarks to the Author):

This manuscript by Antonio J. Capezza et al. reported a process method that using potato protein concentrate as superabsorbent. This work is interesting and also innovative for providing a green method for processing potato protein concentrate and fabricating biodegradable superabsorbents. I would recommend minor revisions before being accepted.

1. In terms of potato protein concentrate processing, what work have other researchers done and what cross-linking agents have been used before?
2. What will happen when the EDTAD content beyond 25wt%?
3. PPC/25ED shows high water absorption, what is its possible mechanism?
4. Generally, protein products are prone to mold and deterioration. Why is PPC/25ED not prone to mold?
5. What will be the difficulties in the industrial application of the reported method?

Response to Review 1

Reviewer #1 (Remarks to the Author):

This paper presents interesting and novel results about the functionalisation of potato protein obtained from agricultural side-streams. Thus, a superabsorbent material has been developed in a sustainable way, presenting a high water uptake capacity and biodegradability. So, I recommend its publication after the revision of the follow minor comments:

1) The definition of the principal objective and the characterisation carried out should be improved in the introduction section. Now, they are difuse. In addition, the results should be ommited in this section.

Answer: We have now clarified in the introduction that the objective of the study was to produce biodegradable superabsorbent materials using chemical functionalisation of PPC (acylation) without using any toxic reagent or cross-linker agent. We have also better defined our intention to test the chemical functionalisation process on the potato fruit juice (containing the potato protein in its soluble form) and removed the mentioned results from the introduction.

The clarifications and enhancement of the objectives are marked in yellow in the last paragraph of the introduction.

2) Is there any hypothesis about why adding 15% EDTA worsens water swelling?

Answer: We believe that the Reviewer refers to the behaviour of the PFJ/25ED/A15 sample. In figure 1d, the PFJ/25ED/A15 sample does indeed have lower absorption than PFJ/25ED/A5 or PFJ/25ED/A10, while absorption increases at a longer autoclaving time in PFJ/25ED/A25 sample. We hypothesise that at an autoclaving time of 15 minutes, the protein has denatured in such a way that it has not yet reached its more insoluble form (obtained for PFJ/25ED/A25), but is more soluble than PFJ/25ED/A5 or PFJ/25ED/A10 (autoclaved for 5 and 10 minutes, respectively). The result was evidenced by the high loss from the teabags for PFJ/25ED/A15 compared to those autoclaved for 5 and 10 minutes (Figure 1d). There is insufficient data to conclude these changes' mechanism due to thermal denaturation without extensive study, so we have chosen not to comment in the manuscript.

3) Line 6: Add the term "respectively" after blood.

Answer: It has been added.

4) Line 18: Change "Counterparts" for "materials".

Answer: It has been changed.

5) Line 23: Change % for wt%.

Answer: We have change % for wt% and checked through the manuscript for similar errors.

6) The microorganisms present in the soil are a key parameter for the degradation of bioplastics. It has been taken in consideration?

Answer: The specific microorganism composition present in the soil used for the degradation test was not taken into consideration. Only the soil conditions, e.g. humidity, carbon/nitrogen content and pH were followed to be able to comply with the ASTM D5988-03 standard. The test's objective was to assess a biodegradation percentage of the materials produced compared to the synthetic superabsorbent material. As pointed by the Reviewer, we consider that a specific study on the effect of the different microorganisms present in the soil during the biodegradation is critical for our ongoing works in the field and will therefore consider it in the future.

7) Have analytical studies been carried out?

Answer: In connection to comment 6 from the Reviewer, no specific analytical studies were carried out regarding the soil's microorganisms. As stated by the standard ASTM D5988-03, only the pH, humidity, and carbon to nitrogen content were monitored, beside the emitted CO₂ for assessing the soil biodegradability of the samples.

We have reflected upon the importance of this analytical study and the specific microorganism action in the biodegradation of these protein-based superabsorbents. Thus, we have now included a short text highlighting the significance of this detailed study for future work (page 10).

We would like to thank Reviewer 1 for the careful reading of the manuscript, particularly for pointing out mistakes in the text and for the input regarding the introduction section. His/her comments and input were greatly appreciated.

Response to Review 2

Reviewer #2 (Remarks to the Author):

This manuscript by Antonio J. Capezza et al. reported a process method that using potato protein concentrate as superabsorbent. This work is interesting and also innovative for providing a green method for processing potato protein concentrate and fabricating biodegradable superabsorbents. I would recommend minor revisions before being accepted.

1. In terms of potato protein concentrate processing, what work have other researchers done and what cross-linking agents have been used before?

Answer: The use of potato protein concentrate processing for the production of bio-based superabsorbents is limited. The significant literature in this application field has been reported in the manuscript. When considering the general use of potato protein concentrate processing, more literature has been reported (although not extensive) in the area of food, bioplastics, and protein nanofibrils. The benefits of using potato protein concentrate processing rely on the fact that additional cross-linkers are not required due to the endogenous cross-links present in the protein. At the same time, glycerol has been reported as a plasticiser in the fabrication of bio-plastics.

We now added a text regarding the use of potato protein processing in other application areas and emphasised that this protein type generally does not require additional cross-linking. This has been adjusted in the introduction, and four new references have been added (page 2, references 26-29 in the revised manuscript). A short clarification regarding the fact that no cross-linkers were used in this work was also added in the introduction.

2. What will happen when the EDTAD content beyond 25wt%?

Answer: We have tested the use of 1:1 EDTAD:protein mass ratio. We showed that the addition of this high amount of EDTAD did not increase the materials' water swelling

capacity. The result is suggested because all the available functional groups in the potato protein been reacted beyond 25 wt% EDTAD addition, which was the concentration showing the maximum water swelling capacity.

We have clarified this remark by the Reviewer in the revised text (page 3) and mentioned the preparation of this sample in the "Methods" section (page 14).

3. PPC/25ED shows high water absorption, what is its possible mechanism?

Answer: The acylation mechanism using EDTAD consist of a ring-opening reaction and condensation reaction of the EDTAD on the lysine-residues (preferentially) on the protein. Damodaran et al. have shown using the trinitrobenzenesulfonic acid (TNBS) method that with increasing the amount of EDTAD added to the protein suspensions, the amount of available lysine groups decreases and the amount of carboxylic acid content in the protein increases (reference 32, 47, 48, and 51 in the manuscript). The increase in carboxylic acid groups has been ascribed to increase the electrostatic repulsion in the protein, thereby promoting higher swelling capacity, which agrees with the results obtained herein. Previous work lead by Damodaran et al. and Cuadri et al. have also shown that concentrations beyond 20 wt% of EDTAD provide better swelling performances in the protein (reference 44, 45, and 46 in the manuscript).

To highlight the reaction mechanism, we have included a clarification in the "Results" section (page 3) and emphasised that the use of concentrations above 25 wt% did not result in any improvement in the material's swelling capacity (in connection with point 3 by the Reviewer).

4. Generally, protein products are prone to mold and deterioration. Why is PPC/25ED not prone to mold?

Answer: Indeed, mould deterioration is already observed at seven days of 100 % relative humidity exposure in the untreated protein (Figure 4a). On the contrary, as pointed by the Reviewer, the sample PPC/25ED and PPC/Ref did not show macroscopic mould deterioration (Figure 4b, Supplementary Figure 5). According to previous studies about functionalised chitosan, it has been demonstrated that mould/bacterial growth inhibition is related to electrostatic interactions and stereo hindrances by the biopolymer in the protein adsorption responsible for cell proliferation.^{1,2} In the study conducted by Hu et al., the sole addition of acrylic acid grafted groups on chitosan substrates (forming negatively charged groups)

decreased the protein adsorption (in $\mu\text{g}/\text{cm}^2$) from 100% to 60%, whilst the bacterial growth was reduced from 140% to 1%.^{2,3} In similar works conducted by Chatelet et al. and Shi et al., the degree of acetylation of chitosan (promoting more carbonyl groups on chitosan) resulted in having a critical role in the cell adhesion and proliferation, where higher acetylation resulted in lower cell adhesion.^{1,4} It has also been suggested that other functional groups, e.g. hydroxyl groups, can produce similar inhibition results. In a more in-depth study performed by Ikeda et al., it was shown that functional groups could associate with anions/cations on the bacterial wall, disrupting the mass transport through the cell walls and promote cell death.⁵ Based on the previous studies and the functionalisation methods applied herein, the alkaline treatment on the proteins (PPC/Ref sample) combined with the EDTAD acylation (producing more charged carboxylic acid groups, PPC/25ED) possibly provides similar effects in terms of mould/bacterial growth inhibition properties of the materials.

We consider the question raised by the Reviewer is important to be shortly reflected in the manuscript. However, we prefer not to speculate further in this area as biological assessment studies were not performed in this study. We have added a short text on page 10, together with five new relevant references (references 62-66 in the revised manuscript), providing a suggestion for this samples' mould inhibition mechanism, emphasising that this is relevant for future works in the area.

5. What will be the difficulties in the industrial application of the reported method?

Answer: We are currently evaluating the applicability of the reported method at an industrial application scale and performing LCA analysis and cost evaluation of the process (to be published in the future). However, considering the technique herein reported, we can conclude that the most challenging factors for industrial scalability will be in the recovery of the unreacted EDTAD from the supernatant and the effect of sample storage in the swelling properties.

We consider that this question is relevant for evaluating this method for the future production of potato protein-based superabsorbents. Thus, we have now clarified the conclusions emphasising the difficulties of the reported method in the industrial application as future work (page 13).

We would like to thank Reviewer 2 for the careful reading of the manuscript. And in particular, the question concerning the mould grow mechanism made us do a literature review on mould inhibition and reflect upon suggested mechanisms for our results. His/her comments and input were greatly appreciated.

1. Chatelet, C.; Damour, O.; Domard, A., Influence of the degree of acetylation on some biological properties of chitosan films. *Biomaterials* **2001**, *22* (3), 261-268.
2. Hu, S.-G.; Jou, C.-H.; Yang, M.-C., Antibacterial and biodegradable properties of polyhydroxyalkanoates grafted with chitosan and chitooligosaccharides via ozone treatment. *J. Appl. Polym. Sci.* **2003**, *88* (12), 2797-2803.
3. Hu, S. G.; Jou, C. H.; Yang, M. C., Protein adsorption, fibroblast activity and antibacterial properties of poly(3-hydroxybutyric acid-co-3-hydroxyvaleric acid) grafted with chitosan and chitooligosaccharide after immobilised with hyaluronic acid. *Biomaterials* **2003**, *24* (16), 2685-2693.
4. Shi, Z.; Neoh, K. G.; Kang, E. T.; Wang, W., Antibacterial and mechanical properties of bone cement impregnated with chitosan nanoparticles. *Biomaterials* **2006**, *27* (11), 2440-2449.
5. Ikeda, T.; Hirayama, H.; Yamaguchi, H.; Tazuke, S.; Watanabe, M., Polycationic biocides with pendant active groups: molecular weight dependence of antibacterial activity. *Antimicrob Agents Chemother* **1986**, *30* (1), 132-136.

REVIEWERS' COMMENTS:

Reviewer #1 (Remarks to the Author):

The authors have contemplate all the comment. So, the manuscript has been highly improved. In this way, I recommend its publication in the present form.

Reviewer #2 (Remarks to the Author):

The author answered the question very well, it is recommended to be accepted.